# Small Caliber Bulletproof Test of Warships’ Hulls

**DOI:** 10.3390/ma13173848

**Published:** 2020-08-31

**Authors:** Radosław Kiciński, Andrzej Kubit

**Affiliations:** 1Mechanical and Electrical Engineering Department, Polish Naval Academy, 81-103 Gdynia, Poland; 2Faculty of Mechanical Engineering and Aeronautics, Rzeszow University of Technology, 35-959 Rzeszów, Poland; akubit@prz.edu.pl

**Keywords:** FEM (finite element method), impact resistance, bullet resistance, fast-changing processes, minehunter

## Abstract

The article presents the characteristics of 1.3964 steel and the results of firing a 7.62 mm projectile with a steel core. A simplified Johnson–Cook material model for steel and projectile was used. Then, a FEM (finite element method) simulation was prepared to calibrate the material constants and boundary conditions necessary to be used in simulations of the entire hull model. It was checked how projectile modeling affects the FEM calculation results. After obtaining the simulation results consistent with the experimental results, using the model of a modern minehunter, the resistance of the ship’s hull to penetration by a small-caliber projectile was tested.

## 1. Introduction

The thickness of the hulls of warships depends on their combat purpose. In extreme cases, it can be from 3 mm for mine warfare ships to even 650 mm on the 1941 battleship “Yamato” [1]. Nowadays, armored units are no longer used, which, along with the development of the quality of the produced steel and changes in the concept of war at sea (reduction of the use of artillery projectiles), has changed the thickness of the ships’ hulls. To reduce the ship’s magnetic field, glass-polyester laminates were also used. These materials, in most cases, have almost no bullet resistance.

In the case of watercraft, another important parameter is the mass above the center of gravity. Increasing the thickness of the superstructure plating increases the height of the center of gravity, which affects the stability of the ship, and therefore they are relatively thin. Taking into account the threat of asymmetric conflicts, there is a possibility of damaging the interior of the ship using small-caliber weapons. Such an attack can occur from a motorboat, other watercraft, or even a drone [2]. An example of a terrorist threat which has recently developed is piracy. In 2008, the supercontainer “Sirius Star” was hijacked in south-eastern waters around 450 miles off the coast of Kenya [3]. The remote location of the attack proves that the pirates have specialized equipment, skills, and better weapons. Increasing acts of piracy and the multiplicity and variety of asymmetric threats require remedial measures. In the case of warships, appropriate patrols may be organized in case of risk. The situation is different for ships for which a private shipowner or oil company usually does not employ trained intervention groups. In each of the cases presented, human life must be protected while serving at sea. Since the threat may occur on any type of vessel, anywhere on the ship, the problem of the ship’s bullet resistance should be considered significant.

The world is looking for solutions that will increase the resistance to penetration by a bullet with a slight increase in weight [3]. Tests are also carried out to determine the bullet resistance of materials [4,5,6,7,8,9]. Due to the change in the concept of war at sea, the plating of both ships and warships has decreased and ranges from 2 to 20 mm in most cases. Therefore, it is worth analyzing how much damage inside the hull can be caused by a 7.62 mm bullet with a steel core. As part of the work [10], the bullet resistance of 1.3964 was tested. It is non-magnetic stainless steel with an austenitic structure, characterized by high strength properties and very good corrosion resistance. The steel has comparable corrosion resistance to other austenitic steels, high yield strength, resistance and stability of mechanical properties at low temperatures, and non-magnetic properties under special operating conditions. The steel is also resistant to intercrystalline corrosion, seawater, salts, chlorides, sulfur compounds, nitric acid, formic acid, and phosphoric acid. This steel is used in the production of chains, parts of heat exchanger installations, bolts, nuts, pump shafts, and springs. It can also be used for the production of fittings, valves, or fragments of critical infrastructure in the chemical, nuclear, marine, shipbuilding, cryogenic, and food industries [11]. Due to its universality and stability of mechanical properties, as well as non-magnetic properties, 1.3964 steel is also used for the production of modern minehunter hulls. This article presents the bullet resistance of 1.3964 steel to penetration by a 7.62 mm steel core bullet. It also presents the finite element method (FEM) simulations of modern minehunter hull structure bullet resistance. The main purpose of the work is to use the microscale experiment for calculations on the macro scale. In addition, the aim of the article is to build awareness of the vulnerability of ships to even the simplest armament, which is a 7.62 mm projectile.

## 2. Materials and Methods

The research began with examining the material characteristics. Material characteristics were obtained by carrying out a static tensile test in accordance with PN-EN ISO 6892-1: 2016-09 [12]. Due to the high strength and hardness of the tested steel, the areas for the samples were cut from its sheet (parallel and perpendicular to the rolling direction) using water jet cutting.

The steel used as the test material was austenitic, stainless steel with the symbol: EN/DIN X2CrNiMnMoNNb 21-16-5-3 [13,14]. Its strength as a construction material depends on the type of treatment that it is subjected to at the metallurgical stage. By using, for example, “cold” hardening, a tensile strength of 700 ÷ 950 MPa can be obtained. Moreover, 1.3964 steel is fully weldable but has restrictive rules regarding the welding technology used and the selection of the chemical composition of the binder (1.3954 or 1.3984). The chemical composition of the tested steel is presented inTable 1. 

Then, based on the data obtained from the static tensile test, following the mathematical description presented in [10,16,17,18], the real material characteristics for the FEM calculations were determined. In CAE (computer aided engineering) programs, entering material data with the use of a table (graph points) is cumbersome and increases computation time due to extrapolation between curve points. This problem can be avoided by using material models. In fast-changing processes, such a material can be described using a simplified Johnson–Cook (J-C) polynomial [19] in the form of Equation (1): (1)$$\sigma =\left(A+B\varepsilon^{n}\right)\cdot \left(1+C\ln\left(\frac{{\dot{\varepsilon }}^{pl}}{{\dot{\varepsilon }}_{0}}\right)\right)$$
where *σ*—stress; *A*—material constant; *B*—hardening parameter; *ε*—plastic strain; *n*—strengthening exponent; ${\dot{\varepsilon }}^{pl}—$. equivalent plastic strain rate; ${\dot{\varepsilon }}_{0}$—reference strain rate.

To determine the material model, the MATLAB environment was used. The program using the so-called engineering method [20] fits the curve by regression to the compiled data, with an accuracy of over 95%. The material characteristics developed in this way are shown in Figure 1.

The next stage of the work was to determine the failure criterion for steel 1.3964. Many failure models have been described in the literature [16,21,22,23,24]. Due to the fact that the J-C viscoelastic model was used to describe the plastic characteristics, this study uses a simplified Johnson-Cook failure model in the form of [25,26] Equation (2): (2)$${\bar{\varepsilon }}_{‘}^{pl}=\left[d_{1}+d_{2}\exp\left(-d_{3}\eta \right)\right]$$
where *d*_1_—strain for which ultimate strength is assumed; *d*_2_, *d*_3_—material constants describing the reduction in material stiffness; *η—*triaxiality.

To determine the value of the failure criterion, tests on the shooting range were carried out. The test consisted of shooting through samples with a thickness of 4, 6, 8 mm. Based on the results, a graph of the drop in velocity of the projectile behind the sample as a function of its thickness was created. The tests were performed on rectangular samples with dimensions of 100 × 140 mm, the measuring part of which was a square with a side equal to 100 mm. The velocity of the projectile was measured before and after the specimen shot through. For this purpose, a Doppler projectile velocity meter and properly prepared short circuit grids were used, which were spaced 1 m apart (Figure 2).

Then, using an oscilloscope, the time difference between the pulses was measured, which made it possible to determine the speed of the bullet in front of and behind the sample. The use of short-circuit grids made it possible to exclude accidental impulse formation caused by, e.g., an acoustic wave moving in front of the projectile. A 7.62 mm projectile was fired from the sniper rifle.

It was assumed that the difference in velocity is the result of a decrease in the total energy absorbed by the material in the form of friction and all accompanying phenomena. The split samples were also shot to check how the empty space affects the decrease in the kinetic energy of the projectile. The measured parameters from the shooting of 1.3964 steel samples are presented in Table 2.

The impact load tasks are very nonlinear. Each FEM task should be treated individually and with caution, because an excessive number of variables may have a negative impact on the results. In the case of bullet penetration, the constitutive equation is more complex due to additional criteria. This equation can be represented as Equation (3): (3)$$M\left(U,n\right)\ddot{U}+C\dot{U}+K\left(U,{\dot{\varepsilon }}_{\mathrm{pl}},\varepsilon_{\mathrm{failure}},n\right)U=F\left(m_{\mathrm{bullet}},\alpha ,v,t,{\dot{\varepsilon }}_{\mathrm{bullet}},\varepsilon_{\mathrm{failure}},C_{\mathrm{int}},\mu \ldots \right)$$
where *K*—structure stiffness matrix; *M*—inertia matrix; *C =* α*M +* β*K*—damping matrix (where α and β are constant coefficients); U,$\;\dot{U}$,$\;\ddot{U}$—displacement, velocity, and acceleration vector; *F*—load vector; ${\dot{\varepsilon }}_{\mathrm{pl}}$—strain rate; ε_failure_—failure strain; *n*—type of material layers and their number; v—projectile speed before impact; *t*—time; α—projectile angle of incidence; $m_{\mathrm{bullet}}$—bullet mass; *C*_int_—interactions and contact forces; *μ*—coefficient of friction.

The above considerations show the multitude of factors influencing the solution of tasks in the field of impact loads. An engineer facing a problem in this area must find a compromise between the accuracy of the solution and the number of factors considered.

With the change in the projectile velocity, the stress and strain values in the sample change. Comparing them with each other is an engineering challenge and depends on many input parameters defined when solving a FEM task. Considering the above, the influence of selected input parameters on the velocity of the projectile behind the sample with a thickness of 4 mm was analyzed. During the calculations, the influence of the following parameters was taken into account:
*A*—material constant of the J-C model;*B*—hardening parameter of the J-C model;*n*—strengthening exponent of the J-C model;*C*—J-C strain rate parameter;*d*_1_, *d*_2_, *d*_3_—J-C failure criterion;*u*_failure_—displacement at failure;*μ*—coefficient of friction. 


To be able to simulate on a macro scale, it was decided to perform simulations on a microscale, relating it to the conducted experiments. The simulation was calibrated using the boundary conditions and the load presented in Figure 3.

During the calibration, the values of the given input parameter were changed by ±50%, and then FEM calculations were performed. Such a procedure allows us to determine the influence of a single parameter on the obtained results. An analysis was made of how the percentage changes in individual parameters affect the percentage change in the final results. In this way, the input parameters for which the projectile velocity after the sample was consistent for the simulation and the experiment were determined. The list of parameters used in the final model is presented in Table 3.

To complete the task as fully nonlinear, a failure model for the projectile itself should also be proposed. Most studies in this area propose to use the projectile as a rigid body to reduce the computation time. The validity of such a solution has been verified. Using the boundary conditions shown in Figure 3, a series of simulations were carried out with the use of elastic, elastic-plastic, and elastic-plastic with the failure criterion material models. Material data for the projectile were obtained from the literature [4] and are presented in Table 4. 

Based on the FEM simulations, it was noticed that the projectile material model has a significant impact both on the shape and morphology of the puncture (Figure 4.) as well as on the velocity of the projectile behind the sample.

The differences in velocity drop indicate the amount of energy that is dissipated by the projectile itself depending on the material model. A projectile modeled as fully elastic transmits much more kinetic energy than a plastically deformable projectile. A projectile with no failure criteria applied has a greater velocity drop than a projectile with the failure criteria applied. These differences result from the contact between FEM elements that should be destroyed (Figure 5).

Taking these observations into account, concerning the conducted experiments, during modeling the tasks in this work, it was decided to model the projectile as fully deformable with the assigned failure criterion.

The next step was to select the material constants of the J-C failure model in such a way that the simulation results correlated with the experiment results as accurately as possible. For this purpose, a model of the sample and the projectile was created, and then a series of simulations were performed, which allowed for the determination of the appropriate material constants. This approach allowed the simulation to be calibrated, which made the results more realistic in further calculations.

A series of simulations was carried out using the results of a 1.3964 steel shot-through. The projectile velocity after the sample and the morphology of deformations from the simulation were visual compared to the actual deformations of the samples. The simulation was prepared as a fully nonlinear problem with the use of the friction coefficient equal to 0.05. The results depended on both the deformation of the test material and the projectile. The sample model was made using three-dimensional 8-node elements. Then, a series of simulations was performed for samples with a thickness of 4, 6, 8 mm, the results of which were related to the experiment. The geometry of the task performed to verify the material constants is shown in Figure 3.

Comparing the simulation results with the experiment, the material constants for the tested steel, necessary for calculations in CAE programs, were determined. The coefficients are presented in Table 3.

Based on the experiment and simulation, the material constants have been determined in such a way as to reflect the behavior of the steel in the case of a small-caliber projectile impact. The differences in the results ranged from 0.15% to 22%, which, in the case of such a nonlinear problem, is an acceptable result.

Using the material constants determined based on experiments and simulations, an analysis of the hull strength of a modern minehunter to shot by a projectile was performed. Using photos and promotional materials [28,29,30], the geometry of the ship’s hull was created (Figure 6).

Due to the confidentiality of information about the ship, a random layout of the cabins inside was proposed. The local hull stiffening was also modeled. Then, a 7.62 mm bullet shot from the stern was simulated to analyze the potential damage inside the ship (Figure 7).

A simulation of a projectile shot into metal sheets with a thickness of 2, 3, and 4 mm was carried out, which made it possible to determine how many ship bulkheads (layers of steel) could penetrate a small-caliber projectile before losing its kinetic energy.

## 3. Results

Based on the conducted tests and simulations, a graph of the resistance of steel 1.3964 to a shot by a 7.62 mm bullet was made (Figure 8).

Based on the chart, it can be concluded that the simulations most accurately reflect the material bullet resistance for the plates with a thickness of 4 mm. The simulation results show a tendency for a linear decrease in speed, while, in the case of the experiment, this line changes its character to become more nonlinear.

Knowing the differences in the results between the experiment and the simulation, a macro-scale simulation of a shot at the ship’s hull was performed. The plating thickness was modified, which made it possible to determine the damage in a given ship compartment depending on the plating thickness. The simulation results are shown in the figures below (Figure 9, Figure 10, Figure 11 and Figure 12).

The simulation shows that the plating thickness has a significant impact on the destruction inside the ship. Furthermore, this article considers the best possible case of hitting each hull stiffener. In the most unfavorable situation (the projectile only hits the bulkhead), there is a probability of penetrating the ship through almost all ship compartments, if it were made of 4 mm thick sheets. It is worth noting that, in the case under consideration, only the 7.62 mm bullet is tested. For larger calibers, the damage would be much more extensive.

## 4. Discussion

This paper presents the contact issue of fully deformable bodies. This required the use of complex FEM operations defining contacts between each of the model elements. Besides the above, the use of failure criteria creates new contacts at each time step. Taking into account the above, it can be assumed that, according to the authors, the differences in the results at the level of 22% are acceptable and allow the model to be used for further calculations.

The model showed a trend for a linear decrease in velocity with increasing sample thickness, which is inconsistent with the experimental results (Figure 8). For this reason, the coefficient of friction must be taken into account for higher material thicknesses. It was found that, for the presented sample thicknesses up to 8 mm, the interaction time is too short and the friction coefficient is not significant. The coefficient of friction for thicker plates shows a significant impact on the results.

The proposed failure criterion is sufficient for the small-caliber projectile penetration calculations; however, it should be borne in mind that the material may deform differently under a different type of load. 

In the finite element modeling, the responses of small deformation and the large deformation are not the same. In the small deformation, the first-order elements are most suitable. In case of large deformation, the first-order element is not enough. During the simulations, it was checked that the use of first-order elements in the presented simulations does not significantly affect the results but extends the calculation time due to the increased number of nodes.

A fully nonlinear task was solved in the work due to the use of the damage model of the tested material and projectile. A review of the literature and studies on this topic proposes the use of a projectile as a perfectly rigid body to reduce the computation time. The choice of the deformable projectile model is more appropriate because the stern of the ship is at an angle of 10 ° to the projectile trajectory. Shot at an angle changes the shape of the projectile [31], which starts to rotate. It causes a reduction in its forward velocity and increases the projectile interaction area with the next obstacle. This has significant consequences in the analysis of the destruction of the ship’s hull (Figure 13).

The article also aims to raise awareness of the bullet resistance of steel structures. Throughout the world, research is constantly being carried out on the implementation of new solutions to increase the bullet resistance of the structure. Nowadays, not only ships are exposed to small-caliber shells. Increasing acts of piracy and terrorism require ship-owners to increase the safety of crew members. For ships with a plating thickness of less than 2 mm, it may not be possible to ensure an adequate level of safety for the crew against sub-machine gunfire.

The presented research is part of the work in which new solutions for ballistic shields (coatings) for effective use in shipbuilding are sought. The work is undertaken in order to develop an innovative cover for the protection of critical cabins on vessels. One of the proposed solutions to the presented problem may be dividing the plating by thickness (split in thickness).

Based on research, simulations, and literature [32,33], it was found that split samples have greater bullet resistance than uniform samples. During the experiment, it was shown that the combination of 6 + 4 mm plates is able to stop a 7.62 mm caliber bullet. This was also demonstrated during the simulation, the results of which are presented in Figure 14 and Figure 15.

The analysis of the simulation of penetrating a 10 mm thick material with a bullet allows us to draw the following conclusions:
The projectile impact on the single sample structure results in the sample puncture. The projectile made a cork that moved with it at a speed of around 200 m/s (Figure 14a);Shooting split samples did not show any breakthrough. The projectile was stopped in the space between the plates (Figure 14b);Dividing the sheets by thickness allowed us to increase its bullet resistance to a level comparable to a sheet with a thickness of 12 mm (Figure 14c).

For simple and cheap solutions, it is worth considering the concept of dividing the plating by thickness to increase bullet resistance.

## Figures and Tables

**Figure 1 materials-13-03848-f001:**
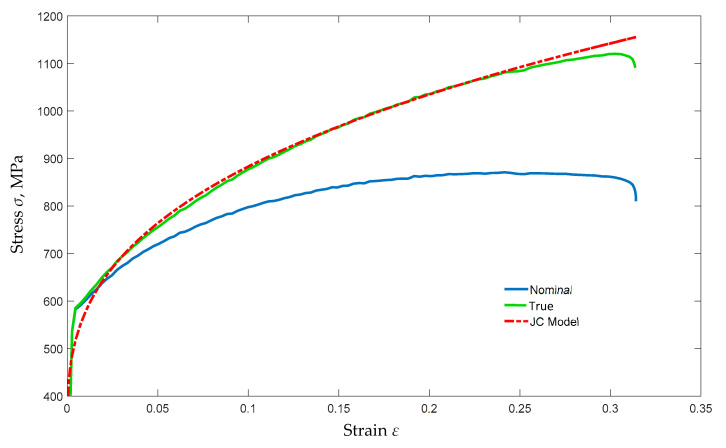
Comparison of the results of a static tensile test and program generated Johnoson-Cook model.

**Figure 2 materials-13-03848-f002:**
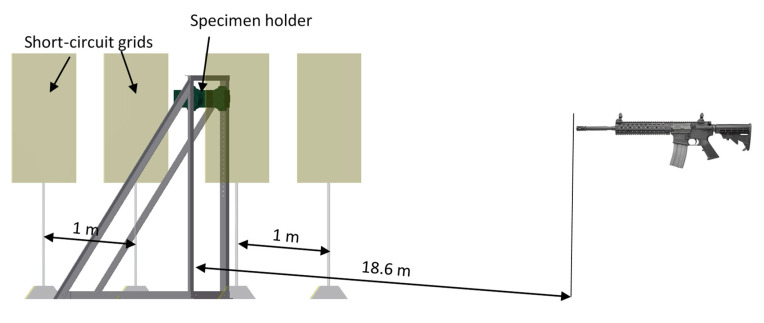
Arrangement of measuring devices in the shooting range.

**Figure 3 materials-13-03848-f003:**
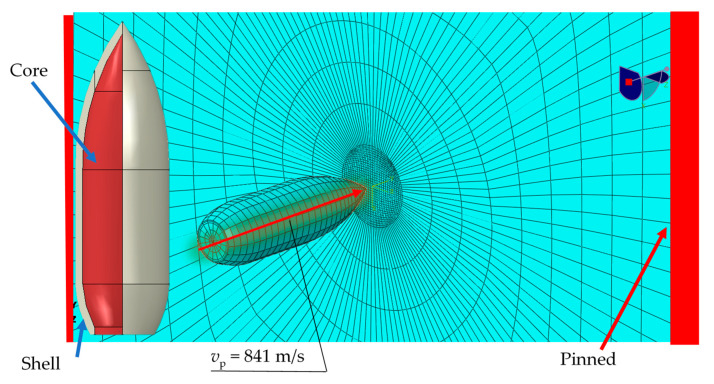
Geometry and discretization of the projectile and the sample [7].

**Figure 4 materials-13-03848-f004:**
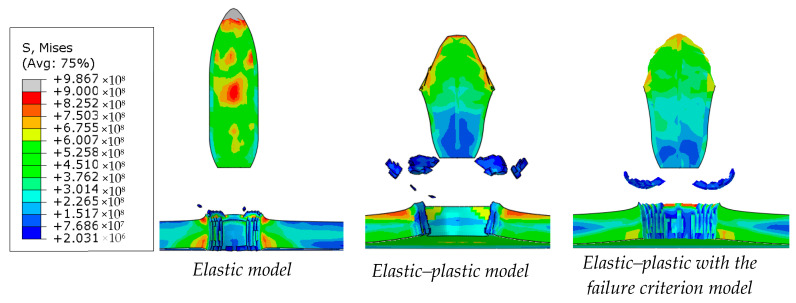
The shape of the projectile and the morphology of the penetration of the sample depending on the used projectile model for a sample with a thickness of 4 mm.

**Figure 5 materials-13-03848-f005:**
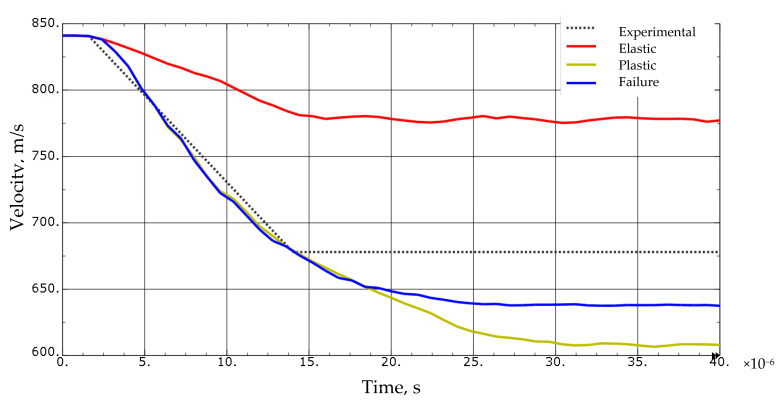
A comparison of projectile velocity drops depending on the projectile model used. Simulation of shooting at a 4 mm thick sample of 1.3964 steel.

**Figure 6 materials-13-03848-f006:**
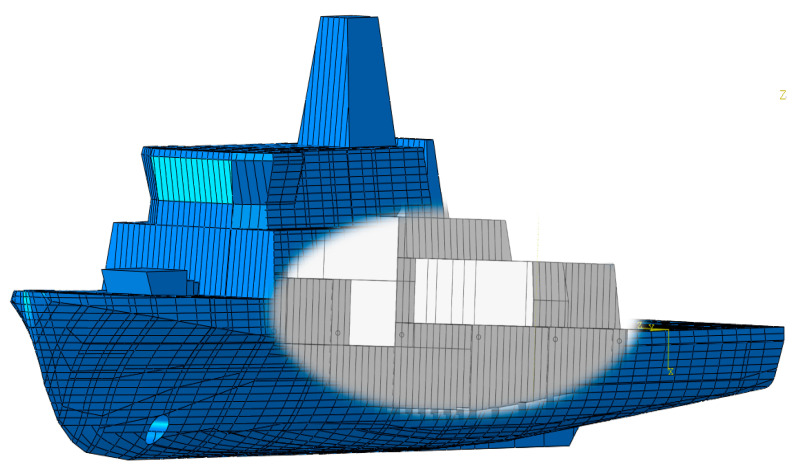
Modern minehunter geometry with random cabin layout.

**Figure 7 materials-13-03848-f007:**
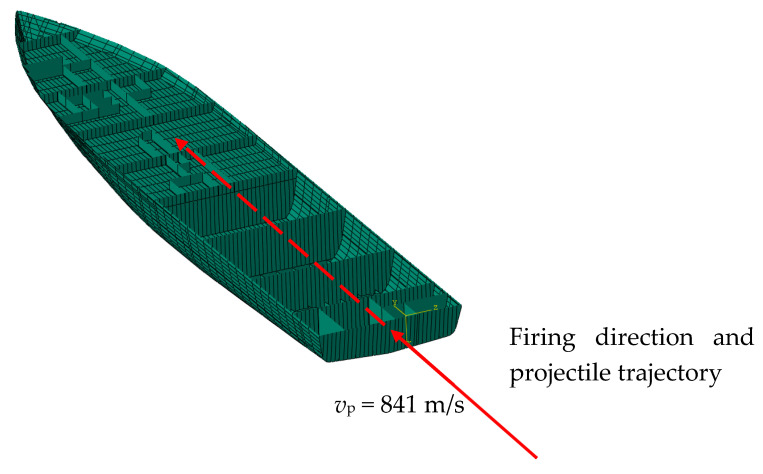
Ship’s interior with the expected projectile trajectory.

**Figure 8 materials-13-03848-f008:**
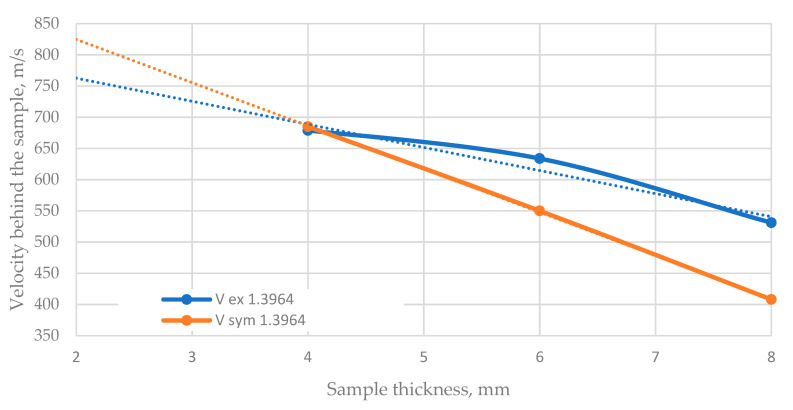
Comparison of projectile velocity results behind the samples of different thicknesses.

**Figure 9 materials-13-03848-f009:**
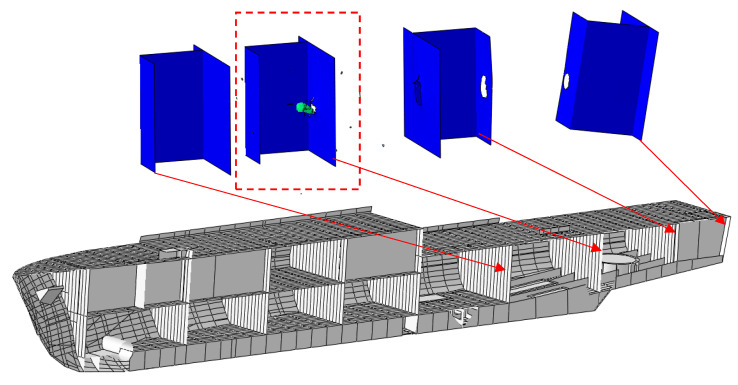
The 2 mm thick hull bullet penetration results.

**Figure 10 materials-13-03848-f010:**
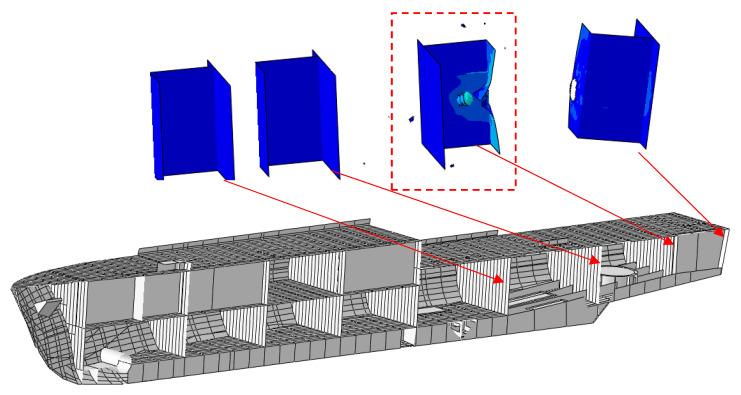
The 3 mm thick hull bullet penetration results.

**Figure 11 materials-13-03848-f011:**
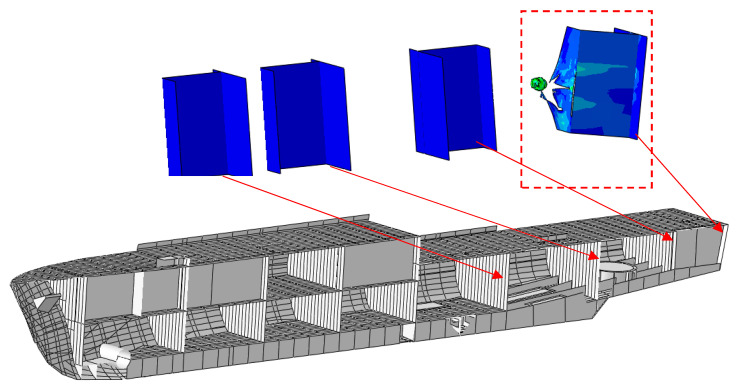
The 4 mm thick hull bullet penetration results.

**Figure 12 materials-13-03848-f012:**
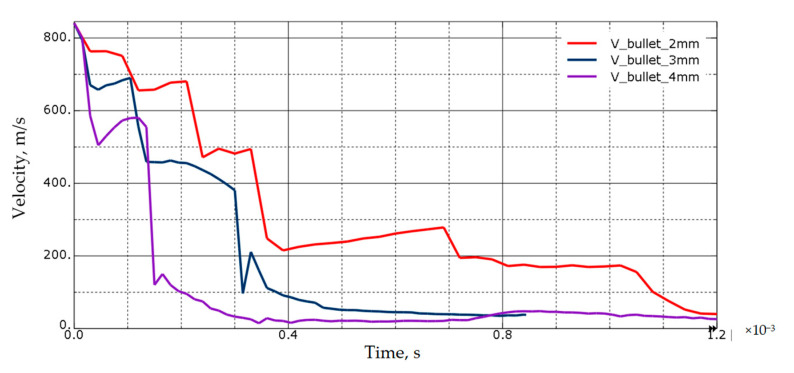
Projectile velocity decrease for different hull plating thicknesses.

**Figure 13 materials-13-03848-f013:**
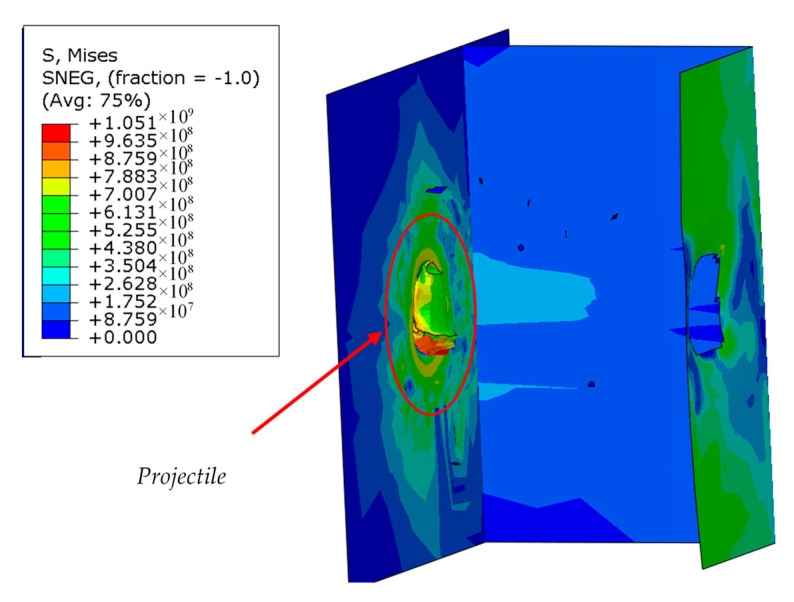
Side hitting projectile due to the torque from the impact at an angle.

**Figure 14 materials-13-03848-f014:**
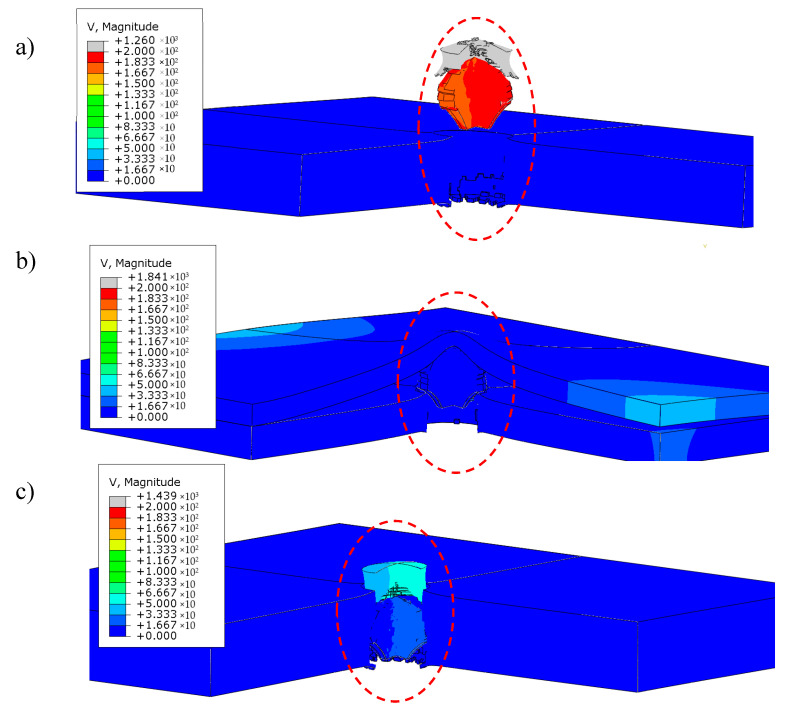
Comparison of FEM results for samples with a thickness of (**a**) 10 mm single, (**b**) 10 mm split, (**c**) 12 mm single.

**Figure 15 materials-13-03848-f015:**
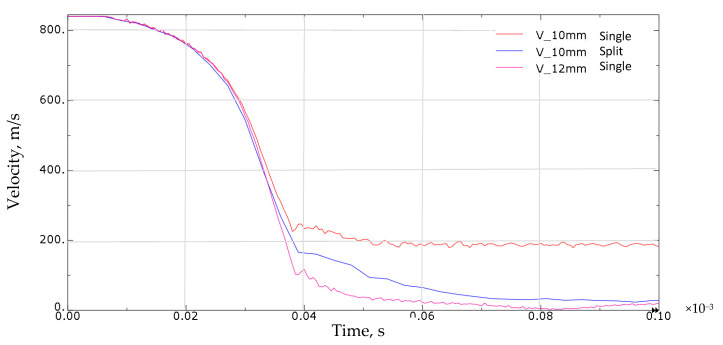
Projectile speed drop. Results of the FEM simulation for samples with a thickness of 10 mm (single and split) and 12 mm (single).

**Table 1 materials-13-03848-t001:** Chemical composition of 1.3964 steel [15].

EN-10088		1.3964/ X2CrNiMoNNb 21-16-5-3	
ASTM		XM-19	
Chemical Composition		Catalog [14]	According to the Metallurgical Certificate
C	%	Max. 0.03	0.012
Mn	%	4÷6	4.42
Cr	%	20÷21.5	20.32
Ni	%	15 ÷17	15.46
Mo	%	3÷3.5	3.15
Nb	%	Max. 0.25	0.12
N	%	<0.11	0.305
Fe	%	residue	residue
Si	%	–	0.36
P + S	%	Max. 0.019 + 0.0004	

**Table 2 materials-13-03848-t002:** Test results for 1.3964 steel.

Sample Thickness mm	Sample Characteristics mm		Gap between Samples	*v*_p_ m/s In Front of the Specimen	*v*_k_ m/s Behind the Specimen	*v*_kaverage_ m/s
4	single	-	None	840.9	683.06	678.91
4	single	-	None	834.8	674.76	
6	single	-	None	842.7	642.67	633.78
6	single	-	None	839.9	626.57	
6	single	-	None	843.7	632.11	
8	sgle	-	None	838	523.29	530.96
8	single	-	None	831	530.22	
8	single	-	None	836.6	539.37	
8	4+4	Gap	20 mm	840	464.25	461.69
8	4+4	Gap	20 mm	841	459.14	
8	4+4	Gap	40 mm	856	456.62	456.00
8	4+4	Gap	40 mm	840	455.37	
8	4+4	Gap	60 mm	842	413.56	417.04
8	4+4	Gap	60 mm	843	420.52	
8	4+4	Gap	80 mm	840	410.17	410.17
8	4+4	Gap	100 mm	835	420.52	420.52
8	4+4	Gap	120 mm	839	448.83	448.83
8	4+4	Gap	140 mm	845	423.73	423.73

**Table 3 materials-13-03848-t003:** Material constants for 1.3964 steel.

Material	Elastic		J-C Plasticity				J-C Failure		
	*E*	*ν*	*A*	*B*	*n*	*C* [27]	*d* _1_	*d* _2_	*d* _3_
1.3964	240 GPa	0.3	302	1250	0.3334	0.006	0.02	0.05	0.5

**Table 4 materials-13-03848-t004:** Material properties for the shell and the bullet core [4].

Part of the Projectile	Elastic		J-C Plasticity				J-C Damage Criterion		
	*E*	*ν*	*A*	*B*	*n*	*C*	*d* _1_	*d* _2_	*d* _3_
Core	210 GPa	0.3	234.4	413.8	0.25	0.0033	5.625	0.3	−7.2
Shell	120 GPa	0.33	448.2	303.4	0.15	0.0033	2.25	0.0005	−3.6

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
