# Peer review of "Small Caliber Bulletproof Test of Warships’ Hulls"

_materials, 2020, doi:10.3390/ma13173848_

Round 1

Reviewer 1 Report

Dear Authors,

I would like to congratulate you for your work presented in manuscript titled "Small Caliber Bulletproof Test of Warships Hulls". I've found the manuscript well written and easy to read. I had a few questions and suggestions for improvement as below:

  1. Does any of the material's properties changes at relatively high strain rates? Normally, JC and JC-damage equations has strain rate components which can be used in FEM studies as it can change how a material behave at high strain rates. As the author's know, strain rate can play an important role in failure.
  2. Please explain the FEM methodology in detail, element type, sizes, boundary conditions and loads used. This is explained briefly for warship's hull studies but definitely not before Fig. 3 where you showed the results of an FEM analysis.
  3. Paragraph started by line 150. How this comparison between experimental and FEA results was carried out? visual? loads? damaged area? please explain in detail.
  4. In discussion, you mentioned about contact for the first time. How did you define the contact and COF in FEM should be mentioned in methodology first. Same as line 207, what did you used as COF of the two material, based on which reference? Also, please provide some supporting data regarding the discussions mentioned in line 200 and 205.

There are a few editorial corrections and areas that I think can be improved. Please check the the manuscript throughly again to ensure it meets journal's requirements. The following are some of the ones I found:

  • Line 23: "ship's Hull's" which probably should be "ships' hulls".
  • Line 66: elongation of calculations", to me elongation is more of a physical change of length, can you find a better word instead?
  • Fig. 1. Legend is not fully shown, no comma is needed before epsilon for "strain" as it's not a unit; use same formatting as of vertical axis or vice-versa.
  • Fig. 4. Please use the same formatting as of the figures (font type and size).
  • There are a few repetitive or fairly closely linked sentences in paragraph starting line 137 and 140. Please remove any repetitive points.

Sincerely

Author Response

Dear reviewer,
Thank you for the informative review.

  1. Of course, the material property values change at different strain rates. The work was guided by an approach from detail to general. An experiment was carried out on samples and then the projectile and material models were selected in a way that best represents reality. Our intention was to use the micro scale experiment for macro scale calculations.
  2. Good point. This will be added to the manuscript. All micro-scale simulations were performed in accordance with the boundary conditions presented in Figure 5. I will rewrite the manuscript to be more readable.
  3. Due to the difficulty of analyzing fast-changing phenomena, it was assumed that the change in kinetic energy (projectile velocity at the front and so much of the sample) is caused by energy dissipation from all accompanying phenomena (deformation, friction, heat, etc.). The projectile velocities in the experiment and simulation were compared. The size of the damage was also compared visually. I will add this to the manuscript.
  4. The influence of input parameters on the results of FEM calculations was analyzed. A series of simulations were performed, in which individual input parameters were changed and the results were compared with the experiment (simulations on a micro scale were calibrated). It was found that for the presented sample thicknesses up to 8 mm, the interaction time is too short and the friction coefficient is not significant. The coefficient of friction for thicker plates shows a significant impact on the results, hence the conclusions given in lines 200-205.

    Due to the many non-linearities of the task, it is difficult to clearly determine whether the quoted conclusions apply to each FEM simulation, so the authors narrowed them down to the projectile-sheet contact simulation only. Drawing broader conclusions requires more experimentation and can be further developed.

  5. I will add editorial and language corrections in the manuscript.

    Thanks again for a very professional review.

Reviewer 2 Report

Reference: Manuscript ID materials-906117

Title: Small caliber bulletproof test of warships hulls

Authors: Radosław Kiciński * , Andrzej Kubit

In this manuscript, the authors presented finite element results of 1.3964 steel and the results of firing a 7.62 mm projectile with a steel core.

Comments:

  • Literature Review and Novelty:

In this manuscript, the literature review and the justification of the problem statement are insufficient to demonstrate the novelty. However, the application domain is important. Otherwise, with the present form, the manuscript is not worthy of publishing in the Materials Journals.

It is strongly recommended to address the problem statement and novelty. Also, the Introduction section needs to be revised.

  • Technical:
  • Some critical issues need to be addressed appropriately:
  1. Why the authors selected specific material and methods? How these are relevant to the problem statement. Also, the authors mentioned, they calibrated model parameters, but they did not explain the calibration methods, which is essential.
  2. About the model: In Figure 4, the authors mentioned three models. But, the authors did not present the governing equations.
  • Conflict in the modeling: In my opinion, the physical process involved in a sudden impact initially localized for a shorter period of time. However, the authors mentioned J-C viscoelastic type model. How are the time-dependent phenomena relevant to the sudden action?
  • Also, the J-C type model is well documented in the literature. Why is further modification essential (Ref: Lines # 9-10)? Interestingly, does Figure 4 represent viscoplastic, or are there any viscoelastic results in this paper?
  • In the Finite Element modeling, the responses of small deformation and the large deformation are not the same. In the small deformation, the first-order elements are well enough. But, in the large deformation (a sudden impact), the first-order element is not enough. Surprisingly, the authors did not mention anything about these critical FE modeling criteria.
  • General: Erroneous Reference lists. Please update them.

Author Response

Dear reviewer,
Thank you for your professional review.

An experiment was carried out on samples and then the projectile and material models were selected in a way that best represents reality. Our intention was to use the micro scale experiment for macro scale calculations. We wanted to present this as a method to other researchers. In addition, our goal is to build awareness of the vulnerability of ships to even the simplest armament, which is a 7.62 mm projectile.
Introduction section has been revised. Hope it is clearer now.

  1. We conducted an experiment on a shooting range and collected the results of testing 1.3964 steel. Then we used them to simulate the bulletproof performance of the entire ship.
    In order to calibrate, the influence of input parameters on the results of FEM calculations was analyzed. A series of simulations were performed, in which individual input parameters were changed and the results were compared with the experiment. Then, based on the results, such input data for the model in the micro scale was used to be able to successfully use them in the macro scale.
  2. Material models refer to the JC equations and are presented as (1) and (2). Additionally, table 3 presents the numerical values adopted for the calculations. I am going to extend Table 3 with Young's modulus and Poisson's ratio.

I don't quite understand. The material model is not time dependent, however the strain and stress values depend on the time step. In large models such as a ship and large distances between contact surfaces (bulkheads), an inadequate time step may result in missing the bulkheads and erroneous results.

Figure 4 relates to the projectile velocity behind the specimen. I compare material models presented in literature (Carbajal, L; Jovicic J; Kuhlmann H Assault Riffle Bullet-Experimental Characterization and Computer (FE) Modeling.)
I have not conducted my own experimental analysis of projectile deformation. I only showed the influence of using a more complex model (than rigid body) on the calculation results.

The modification of the J-C model results from the calibration of the simulation. The use of the strain rate dependent model has a slight influence on the simulation results, and causes the extension of the calculation time.

Also, no differences were shown with the use of second-order elements as the authors did not come up with the idea to demonstrate this in this manuscript, we appreciate your suggestions and will add information on this. The use of second-order elements slightly affects the presented results, but extends the calculation time.

References have been corrected in the manuscript.

Thank you very much for the top-notch review.

Round 2

Reviewer 2 Report

Accepted in the present form.